

# Flexible ureteroscopy for renal stone without preoperative ureteral stenting shows good prognosis

Jiaqiao Zhang[1], Chuou Xu[2], Deng He[1], Yuchao Lu[1], Henglong Hu[1], Baolong Qin[1], Yufeng Wang[1], Qing Wang[1], Cong Li[1], Shaogang Wang[1] and Jihong Liu[1]

[1] Department and Institute of Urology, Tongji Hospital, Tongji Medical College, Huazhong University of Science and Technology, Wuhan, People's Republic of China
[2] Department of Radiology, Tongji Hospital, Tongji Medical College, Huazhong University of Science and Technology, Wuhan, People's Republic of China

## ABSTRACT

**Purpose:** To clarify the outcome of flexible ureteroscopy (fURS) for management of renal calculi without preoperative stenting.

**Methods:** A total of 171 patients who received 176 fURS procedures for unilateral renal stones were reviewed. All procedures were divided into two groups depending on whether they received ureteral stenting preoperatively. Baseline characteristics of patients, stone burden, operation time, stone-free rates, and complications were compared between both groups.

**Results:** Successful primary access to the renal pelvis was achieved in 104 of 114 (91.2%) patients without preoperative stenting, while all procedures with preoperative stenting (n = 62) were successfully performed. A total of 156 procedures were included for further data analysis (56 procedures in stenting group and 100 in non-stenting group). No significant differences was found regardless of a preoperative stent placement in terms of stone-free rate (73.2% with stenting vs. 71.0% without, $P = 0.854$), operative time (70.4 ± 32.8 with stenting vs. 70.2 ± 32.1 without, $P = 0.969$).

**Conclusions:** fURS for management of renal stone without preoperative ureteral stenting are associated with well outcome in short term follow-up. Our study may help patients and doctors to decide if an optional stent is placed or not.

## INTRODUCTION

Improvement of the instruments as well as endoscopic technology has made flexible ureteroscopy (fURS) as an increasingly popular treatment option for patients with renal stones (*Ghani & Wolf, 2015*; *Dauw et al., 2015*; *Wang & Preminger, 2011*). Indication for fURS also has been extended, even for stone larger than 2 cm (*Aboumarzouk et al., 2012*; *Breda et al., 2008*). Preoperative stenting is frequently used to allow passive ureteral dilatation, which is supposed to facilitate the passage of a flexible ureteroscope or ureteral access sheath (UAS). In a study, double-J stent inserted 5–10 days before fURS was

Corresponding author
Shaogang Wang,
sgwangtjm@163.com

recommended as a part of standardized technique to achieve superior results for treatment of renal stones (*Miernik et al., 2012*). However, a preoperative stenting inevitably leads extra cost and time, as well as complications such as flank pain, sexual dysfunction, bothersome urinary symptoms and potential urinary tract infection (UTI) (*Joshi et al., 2003*; *Joshi et al., 2001*; *Shigemura et al., 2012*). Then, patient compliance is an issue for urologists. In addition, urologists have better skill and more confidence to successfully perform fURS procedures for patients without preoperative stenting with increased experience. For the above reasons, more and more fURS procedures without preoperative stenting were carried out in our department.

However, the outcome of Flexible Ureteroscopic Treatment for renal stones without preoperative stenting is undefined. Hence, we reviewed a series of cases to seek the answer.

## MATERIAL AND METHODS

This study was approved by the ethics committee of Tongji Hospital, Tongji Medical College, Huazhong University of Science and Technology. All unilateral fURS procedures from between June 2013 and May 2015 were reviewed. All patients included underwent fURS as initial attempt or substitute for previous failed Extracorporal shock wave lithotripsy (ESWL) or Percutaneous nephrolithotomy (PCNL) procedures to treat solitary as well as multiple renal calculi. The patients with the presence of ipsilateral ureteral stones were excluded. All patients were classified into two groups depending on whether they received a preoperative stent. The patients in stenting group received preoperative stenting (5~8 F) for persistent renal colic, fever, insufficient renal function, or just to dilate the ureter to facilitate subsequent fURS. The patients in non-stenting group had not received stenting before fURS.

All procedures were performed by experienced urologists in our department according to standard operative protocols. UTI was controlled before all operations. For patients with preoperative stenting, firstly the stents was removed through ureteroscopy (URS) or cystoscopy. For patients without preoperative stenting, a rigid URS was performed to detect the whole ureter at first. Then a guidewire was inserted into ureter and UAS (12/14 F, Cook) was placed routinely. A 7.5 F flexible ureteroscope (Storz, Germany) with a 200 $\mu$m laser fibre was used for laser lithotripsy. Nitinol stone retrieval baskets were also used if necessary at surgeon discretion. All patients received ureteral stent at the end of operations. Then all patients underwent KUB or non-contrast CT about four weeks and three months after treatment, and absence of any stone or residual stone fragment $\leq$ 4 mm was considered as stone-free.

We retrospectively recorded data from patient documentation, radiologic images (including X-ray, IVU, and computed tomography), reports of anesthesia and operation. All radiologic images were checked collaboratively by an urologist and a radiologist. Number, location and linear diameter of stones visible in radiologic images were noted in detail. Fever was defined as postoperative body temperature $\geq$ 38. The procedures were performed by a total of five urologists.

Statistical analysis was performed by Statistical Package for the Social science version 13.0. Continuous variables with normal distribution and without normal

distribution were performed with Student's *t*-test and Mann-Whitney U-test, respectively. Categorical variables were performed with chi-squared test or Fisher's exact test. Potential factors associated with stone-free rate including BMI (kg/m$^2$), stone size (mm), stone number, presence of preoperative stenting, type of anesthesia and presence of hydronephrosis were analyzed by multivariate logistic regression analysis. A *P* value of < 0 .05 was considered statistically significant.

## RESULTS

A total of 176 unilateral procedures of fURS from 171 individual patients were included. Figure 1 shows the clinical outcome of all patients. The majority (104 of the 114) of the patients without a preoperative stent were successfully performed fURS, while the left 10 patients underwent a failed attempt for access to the renal pelvis was not achieved. Of the 10 patients, five patients received a stent placement and subsquent fURS, four patients underwent a conversion to PCNL procedure,and one patient received a stent followed by ESWL treatment. All of 62 patients with a preoperative doule-J stent (including the above mentioned five patients who received a stentin first failed attempt) were successfully performed fURS (Fig. 1).

As Fig. 1 showed, a total of 166 procedures successfully entered into renal pelvis to perform fURS lithotripsy. Of the 166 procedures, ten procedures with incomplete data were not enrolled for further data analysis. There were no statistically significant differences between preoperative stenting group (n = 56) and non-stenting group (n = 100) in terms of age, gender, side, BMI, presence of hydronephrosis, type of anesthesia, operation time. Stone number, stone site, and stone burden were also similar in both groups. Positive rate of preoperative urine culture was higher in stenting group. Stone-free rate was similar in both groups. Procedures for solitary stone accounted for 71.2% (111 of 156). For single stone, stone-free rate was highest for stone in renal pelvis, and lowest in lower pole. Stone-free rates were also similar for patients with solitary stone in both groups (Table 1). Multivariate assessment revealed stone size rather than preoperative stenting was the independent predictor of stone-free rate after fURS in this study (Table 2).

Complication rates were low in both groups except for fever. Surprisingly, up to 17.9% of patients with preoperative stent experienced fever postoperatively. Patients without preoperative stenting had a much lower incidence of fever compared with that (6 vs. 17.9%, *P* = 0.027). Perforations of renal pelvis or ureter were more common for patients without a preoperative stent, and were low in both groups. All perforations were not severe, and treated with a postoperative stent. No complications of ureter avulsion occurred in both groups. Severe complications such as bleeding with transfusion or urosepsis were rare (Table 3).

## DISCUSSION

As one of common available treatment options, fURS has occupied more and more positions formerly held by PCNL and ESWL. However, limited data have been referred to the role of preoperative stenting in the treatment of renal calculi by fURS

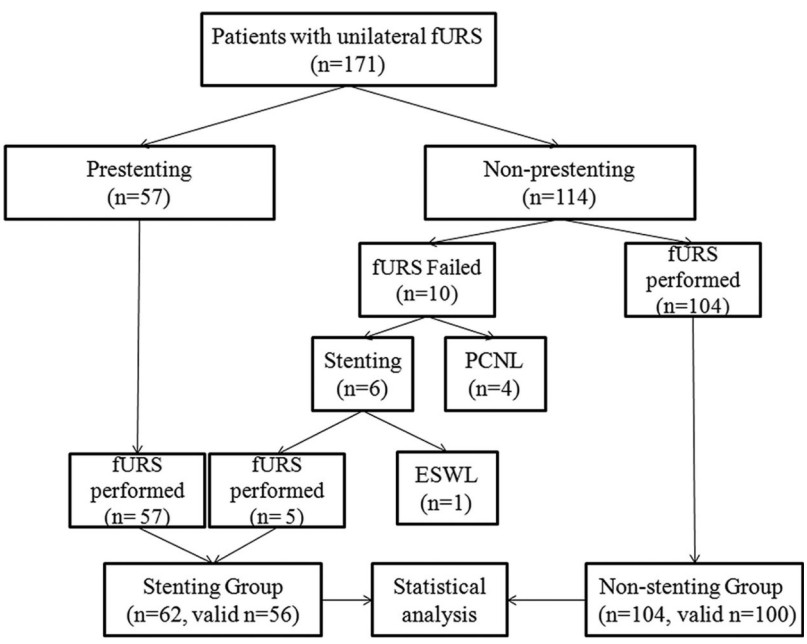

**Figure 1 Flow diagram and patient disposition.** fURS, flexible ureteroscopy; ESWL, Extracorporal shock wave lithotripsy; PCNL, Percutaneous nephrolithotomy.

**Table 1 Patient characteristics in stenting and non-stenting group.**

|  | Stenting group | Non-stenting group | P |
|---|---|---|---|
| Procedures | 56 | 100 |  |
| Age (year) | 51.4 ± 12.8 | 47.6 ± 13.2 | 0.080 |
| Gender (Men/Women) | 32/24 | 62/38 | 0.610 |
| Side (Right/Left) | 24/32 | 47/53 | 0.738 |
| BMI (kg/m$^2$) | 24.2 ± 3.6 | 23.7 ± 3.1 | 0.381 |
| Hydronephrosis (±) | 20/36 | 39/61 | 0.733 |
| Urine culture (±) | 13/43 | 7/93 | 0.006 |
| Anesthesia (Epidural/General) | 30/26 | 48/52 | 0.617 |
| Operation time (min) | 70.4 ± 32.8 | 70.2 ± 32.1 | 0.969 |
| Stone size (mm) | 18.1 ± 8.1 | 18.4 ± 7.7 | 0.812 |
| <10 | 7 | 7 |  |
| 10–20 | 29 | 63 |  |
| >20 | 20 | 30 |  |
| Solitary/multiple stone | 40/16 | 71/29 | 0.955 |
| Solitary stone location |  |  |  |
| Upper pole | 3 | 6 |  |
| Middle pole | 5 | 6 |  |
| Lower pole | 15 | 29 |  |
| Renal pelvis | 17 | 30 |  |
| Stone-free rate | 73.2% (41/56) | 71.0% (71/100) | 0.854 |
| For solitary stone | 85.0% (34/40) | 77.5% (55/71) | 0.458 |

**Table 2 Result of multiple logistic regression analysis to determine factors associated with stone-free rate.**

| Variable | Category | OR | 95% CI | *P* value |
|---|---|---|---|---|
| BMI (kg/m$^2$) | | 1.027 | 0.899–1.173 | 0.699 |
| Stone size (mm) | | 0.407 | 0.206–0.807 | 0.010 |
| Stone number | 1 vs. ≥2 | 0.423 | 0.147–1.216 | 0.110 |
| Preoperative stenting | Present vs. absent | 0.799 | 0.333–1.919 | 0.616 |
| Anesthesia | General vs. epi | 0.907 | 0.392–2.101 | 0.820 |
| Hydronephrosis | Absent vs. present | 1.070 | 0.448–2.558 | 0.879 |

**Table 3 Comparisons of postoperative complications between both groups according to the Clavien-Dindo classification.**

| | Clavien | Stenting group | Non-stenting group | *P* |
|---|---|---|---|---|
| Fever | (Grade II) | 10 (17.9%) | 6 (6%) | 0.027 |
| Renal pelvis/ureter perforation | (Grade IIIb) | 1 (1.8%) | 4 (4%) | |
| Transfusion | (Grade II) | 1 (1.8%) | 0 | |
| Urosepsis | (Grade IVb) | 1 (1.8%) | 1 (1%) | |

(*Lumma et al., 2013*; *Shields et al., 2009*; *Netsch et al., 2012*; *Kawahara et al., 2012*; *Rubenstein et al., 2007*). *Shields et al. (2009)* reviewed a cohort of patients undergoing URS for upper urinary tract calculi and concluded that preoperative stent was positively associated with success, however, without statistical significance. *Lumma et al. (2013)* analyzed data of 550 ureterorenoscopies treated for stones in ureter and renal pelvis. Data indicated that patients with preoperative stent had improved results in stone-free rate and complication rate apart from distal ureteral stones. *Kawahara et al. (2012)* analyzed 51 patients with large (>15 mm) stone, and concluded that URS success rate was higher in the stenting group and may improve SFR. However, most studies were referred to patients with ureteral and renal calculi, and instruments and methods applied for ureteral calculi are quite different from renal calculi. In addition, fURS occasionally will not pass beyond the ureter to reach renal pelvis, which have been mentioned limitedly (*Ambani et al., 2013*; *Mahajan et al., 2009*; *Cetti, Biers & Keoghane, 2011*). Therefore, the aim of this study was to systematically clarify the surgical outcome of fURS without preoperative stenting for renal calculi.

Of the 114 patients without preoperative stenting, 10 patients underwent a failured procedure for access to the renal pelvis was not achieved. In these patients, a narrow ureteric lumen or tortuous ureter caused the failure of access, even though no evidence of obstruction in ureter was found preoperatively or intraoperatively. Half of the 10 patients received a ureteral stent for a period of time (7~40 d) and underwent a second attempt in which narrow ureteric lumen or tortuous ureter disappeared. Obviously, the preoperative stenting passively dilated the ureter and facilitated access to renal pelvis in them. A study by *Ambani et al. (2013)* retrospectively analyzed 41 patients who underwent ureteral stenting after an initial failed attempt of fURS. The second fURS was performed in these patients after 4–34 days and succeeded in 38 patients (93%). Therefore, when a difficult ureter was faced in first attempt of fURS, placement of a stent

and subsequent second attempt was optional and even encouraged. In our study, stone-free rate (three months after operation) in patients with preoperative stenting was higher compared to patients without preoperative stenting, however without statistical significance (73.2 vs. 71.0%, $P = 0.854$). This result was not consistent with some previously published data (*Lumma et al., 2013*; *Shields et al., 2009*; *Chu, Sternberg & Averch, 2011*). One possible reason was that ureteral stones account for a high proportion of all stones in these studies. Multivariate assessment also revealed that preoperative stent was not the independent predictor of stone-free rate after fURS in our study which was similar with a previous study (*Ito et al., 2015*). In our opinion, once the fURS successfully arrived at renal pelvis, stone-free rate for renal calculi would be more likely affected by stone characteristics and pelvicalyceal anatomy rather than the existence of preoperative stenting according to our experience and some published data (*Skolarikos et al., 2015*; *Jessen et al., 2014*; *Resorlu et al., 2012*).

Stone-free rate was not very high in our study. One reason may be that stone size (18.4 ± 7.7 mm) in our study was higher than many studies (*Miernik et al., 2012*; *Perlmutter et al., 2008*; *Hussain et al., 2011*). Although non-contrast CT was the most accurate modality for follow-up of urolithiasis, we also performed KUB for follow-up for its advantages of less cost and radiation exposure.

The operation time was similar in both groups. However, different results had been reported. *Lumma et al. (2013)* found that the operation time in patients with preoperative stent was extended by 4.9 min compared to patients without stent, and attributed this to stent extraction. Another study indicated that preoperative stenting decreased operative time of URS for stones (mainly in the ureter). The possible reason may be that preoperative stenting dilated the ureter and allowed a bigger UAS and better irrigation (*Chu, Sternberg & Averch, 2011*). In our experience, it would consume some time to remove preexisting stents; however, a procedure of rigid URS was generally not necessary before placing the UAS in these patients.

Severe complications were rare in our study, however, postoperative fever rate was much higher compared with other complications. Also, patients with preoperative stenting had a higher incidence of fever compared with without (17.9 vs. 6.0%, $P = 0.027$). The reason may be that majority of patients with preoperative stenting in this study received a stent just because of existence of fever or urinary tract obstruction and patients in stenting group indeed had a significant higher rate of positive urine culture. Therefore, we could not conclude that preoperative stenting increased postoperative fever rate, although an extended time of stenting may cause potential UTI (*Shigemura et al., 2012*). Also, the incidence of fever in our patients without preoperative stenting was similar with the results (3.6% for 10–20 mm renal stones) of CROES URS Global Study (*Skolarikos et al., 2015*).

Perforations of ureter or pelvis were not very common in both groups (1.8% with stent vs. 4% without). As many mild injuries may be ignored in records of our operations, it was hard to know accurate incidence of ureter injuries for all patients. We speculated that ureter injuries (especially for mild injuries) were common according to our experience as well as published data (*Traxer & Thomas, 2013*). However, severe

ureter injuries such as ureter avulsion never happened. In our experience, if unreasonable resistance was perceived when passing through the ureter, it was better to terminate the fURS procedure rather than try with a greater force to avoid severe ureter injury. As many above mentioned substitutes could be selected subsequently, an "adventure" was not necessary.

A recent published large scale study included 1,622 patients with renal stones who were treated with fURS and concluded that preoperative stent increased stone-free rates (79.6% with stent vs. 72.9% without stent) and decreased intraoperative complications (*Assimos et al., 2015*). The results of this large scale study are very valuable, however, we still should note that heterogeneity of reasons for preoperative stenting may affect outcome of fURS for renal stones in different studies. This large scale study indicated that non-stenting fURS for renal stones may have some disadvantages, however, it still had an acceptable prognosis. In addition, it should be emphasized that non-stenting procedures would benefit patients for less cost and suffering.

Our study has some limitations. First, it is a retrospective study from single center. Second, the complications mentioned in our study were early stage complications, while late complications such as ureter stricture were not concerned due to limited follow-up time. Despite these facts, our study is one of the largest series to determine the prognosis of non-stenting procedure on management of renal calculi with fURS.

Our study showed that majority (91.2%, 104 of the 114) of the patients without preoperative stenting were successfully performed fURS at first attempt. The procedures of non-stenting fURS for renal stones have an acceptable prognosis compared with the stenting group in our study. If a preoperative stenting is optional, our study may help the patients and clinicians make a final decision. Large prospective randomized controlled studies are required to further figure out the role of pre-stent, especially for optional cases.

### Funding
This work was supported by National Natural Science Foundation of China (81270787, 81570631). The funders had no role in study design, data collection and analysis, decision to publish, or preparation of the manuscript.

### Grant Disclosures
The following grant information was disclosed by the authors:
National Natural Science Foundation of China: 81270787, 81570631.

### Competing Interests
The authors declare that they have no competing interests.

### Author Contributions
- Jiaqiao Zhang conceived and designed the experiments, performed the experiments, analyzed the data, contributed reagents/materials/analysis tools, wrote the paper, prepared figures and/or tables, reviewed drafts of the paper.

- Chuou Xu performed the experiments, reviewed drafts of the paper.
- Deng He performed the experiments, reviewed drafts of the paper.
- Yuchao Lu analyzed the data, contributed reagents/materials/analysis tools, reviewed drafts of the paper.
- Henglong Hu conceived and designed the experiments, reviewed drafts of the paper.
- Baolong Qin performed the experiments, reviewed drafts of the paper.
- Yufeng Wang performed the experiments, reviewed drafts of the paper.
- Qing Wang performed the experiments, reviewed drafts of the paper.
- Cong Li performed the experiments, reviewed drafts of the paper.
- Shaogang Wang conceived and designed the experiments, reviewed drafts of the paper.
- Jihong Liu conceived and designed the experiments, reviewed drafts of the paper.

### Human Ethics

The following information was supplied relating to ethical approvals (i.e., approving body and any reference numbers):

Ethics committee of Tongji Hospital, Tongji Medical College, Huazhong University of Science and Technology.

IRB ID: 20150404.

### Data Deposition

Xu, Chuou (2016): fURS-rawdata.xlsx. figshare.
https://dx.doi.org/10.6084/m9.figshare.3407122.v1;

Zhang, Jiaqiao (2016): 110 cases unstented cases. figshare.
https://dx.doi.org/10.6084/m9.figshare.3976275.v1.

### Supplemental Information

Supplemental information for this article can be found online at http://dx.doi.org/10.7717/peerj.2728#supplemental-information.

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
