# Peer review of "Flexible ureteroscopy for renal stone without preoperative ureteral stenting shows good prognosis"

_PeerJ, doi:10.7717/peerj.2728_

## Round 0.1 · original submission · Major Revisions

Please clarify what the source/predictor of failure for primary URS in the unstented population is? Are there any predictors?

Reviewer 1 ·

Basic reporting

The tables characterizing the study group are marginal, i.e not very thorough.

There are multiple syntax errors that require correction.

The methods are not described in enough detail.

Experimental design

The methodology is not very scientifically rigorous. Detail on nature of the clinical interventions is lacking.

The statistical analysis appears reasonable.

Validity of the findings

The data is somewhat sparse; there are other factors that can influence the results that are not presented or well delineated.

The findings are not new and are consistent with prior literature.

Additional comments

The authors assess ureteroscopy for renal stone without preoperative ureteral stenting shows good prognosis. Baseline characteristics of patients, stone burden, operation time, stone free rates, and complications were compared between both groups. They conclude that flexible ureteroscopy for management of renal stones without preoperative ureteral stenting is associated with good outcome in short term follow-up.

There are some minor syntax errors which should be corrected.

“The patients in stenting group received preoperative stenting (5~8F) for persistent renal colic, fever, insufficient renal function, or just to dilate the ureter to facilitate subsequent fURS. The patients in non-stenting group had not received stenting before fURS.” The groups are not equal, i.e patients were selectively stented. Moreover, why did they just dilate the ureter with a stent if they thought primary URS was possible (i.e in cases of no infection). From what I can discern from the methodology, this is not a randomized, controlled, trial.

Did any of the patients in study have prior passed stones or stents?

Ureteral perforation should be rare; how did this happen? What was the source of failure for those without stents at time of primary URS (location?)

Other than semirigid URS, was any other ureteral dilation performed, were ureteral access sheaths used.

Why did a patient undergoing URS receive blood transfusion?

The key questions is what is the source/predictor of failure for primary URS in the unstented population? Are there predictors?

Were stones dusted or fragmented/basket? Was it the choice of the surgeon?

How long were the stents indwelling? Does this make a difference?

---

## Round 0.2 · accepted · Accept

The topic of this study has an important impact in patient management.